# Complementary information derived from CRISPR Cas9 mediated gene deletion and suppression

Joseph Rosenbluh[1,2,†,*], Han Xu[1,†,*], William Harrington[1], Stanley Gill[1,2], Xiaoxing Wang[1,2], Francisca Vazquez[1,2], David E. Root[1], Aviad Tsherniak[1] & William C. Hahn[1,2]

CRISPR-Cas9 provides the means to perform genome editing and facilitates loss-of-function screens. However, we and others demonstrated that expression of the Cas9 endonuclease induces a gene-independent response that correlates with the number of target sequences in the genome. An alternative approach to suppressing gene expression is to block transcription using a catalytically inactive Cas9 (dCas9). Here we directly compare genome editing by CRISPR-Cas9 (cutting, CRISPRc) and gene suppression using KRAB-dCas9 (CRISPRi) in loss-of-function screens to identify cell essential genes. CRISPRc identified 98% of previously defined cell essential genes. After optimizing library construction by analysing transcriptional start sites (TSS), CRISRPi identified 92% of core cell essential genes and did not show a bias to regions involved in copy number alterations. However, bidirectional promoters scored as false positives in CRISRPi. We conclude that CRISPRc and CRISPRi have different off-target effects and combining these approaches provides complementary information in loss-of-function genetic screens.

[1] Broad Institute of Harvard and MIT, 415 Main Street, Cambridge, Massachusetts 02142, USA. [2] Department of Medical Oncology, Dana-Farber Cancer Institute, Harvard Medical School, 450 Brookline Avenue, Boston, Massachusetts 02215, USA. † Present address(es): Department of Biochemistry and Molecular Biology, Monash University, 23 Innovation Walk, Clayton, Victoria 3800, Australia and The Hudson Institute of Medical Research, 27-31 Wright Street, Clayton, Victoria 3168, Australia (J.R.); Department of Epigenetics and Molecular Carcinogenesis, The University of Texas M.D. Anderson Cancer Center, Science Park, 1808 Park Road 1C, Smithville, Texas 78957, USA (H.X.). * These authors contributed equally to this work. Correspondence and requests for materials should be addressed to W.C.H. (email: William_Hahn@dfci.harvard.edu).

Loss-of-function genetic screens permit the systematic study of gene function. Until recently, these studies were limited to RNA interference (RNAi)-based technologies and have been complicated by microRNA (miRNA)-like off-target effects. The recognition that CRISPR-Cas9 can be used as a highly specific approach to edit genes now permits one to create gene deletions both for individual genes[1,2] and in screens[3,4]. However, we[5] and others[6] have recently reported a gene-independent proliferation arrest induced by Cas9-mediated DNA cleavage of amplified genomic regions, suggesting a systematic off-target effect in CRISPR-Cas9 screens. Furthermore, studies have demonstrated that single-nucleotide mismatches contribute to Cas9 off-target effects[7].

Another approach for CRISPR-mediated gene suppression is to utilize the ability of Cas9 to recognize specific DNA sequences to target a catalytically inactive Cas9 (dCas9) fused to a transcription repressor domain (KRAB) to transcription start sites (TSS), thereby interfering with gene transcription (CRISPRi[8–10]). Here we evaluate the performance of these two approaches in loss-of-function screens. We find that both CRISPRc and CRISPRi allow one to identify cell essential genes; however, each of these approaches has different off-target effects. Specifically, targeting amplified genes with CRISPRc induces a gene-independent proliferation arrest, while the use of CRISPRi at bidirectional promoters can result off-target effects. We conclude that these two approaches provide complementary information in loss-of-function screens.

## Results

**KRAB-dCas9 proliferation screens with a tiling library.** We cloned Cas9 and KRAB-dCas9 into a lentivirally delivered, blasticidin selectable expression vector (pLX311)[11], and transduced these vectors into four cancer cell lines (HT29, A549, MIAPACA2 and A375) to stably express these two versions of Cas9. To compare the phenotypes induced by KRAB-dCas9 and Cas9, we constructed a lentivirally delivered tiling-pooled single guide RNA (sgRNA) library targeting 7 core essential genes, 21 cancer-related oncogenes and 5 tumour suppressor genes (Supplementary Data 1 and 2). In addition, we included sgRNAs-targeting *HPRT1*, a gene that upon suppression confers resistance to 6-thioguanin (6TG), as well as 979 negative controls (sgRNAs that do not map to any sequence in the human genome). For each gene, we designed sgRNAs tiling the TSS (50 sgRNAs) or exons (50 sgRNAs) (Fig. 1a), to create a library of 4,928 sgRNAs. Following transduction, proliferation changes induced by individual sgRNAs in the presence (1 cell line, Supplementary Data 1) or absence (4 cell lines, Supplementary Data 2) of 6TG were measured 21 days post infection (DPI) using massively parallel sequencing (Fig. 1a).

As expected, 6TG induced a proliferation arrest that was rescued by sgRNAs-targeting *HPRT1* exons 1–9 in cells expressing Cas9 and by sgRNAs targeting the *HPRT1* TSS in KRAB-dCas9-expressing cells (Fig. 1b and Supplementary Fig. 1a). We confirmed that expression of KRAB-dCas9 from pLX311 or pHR[8] induced similar proliferation changes (Fig. 1a). We also found that sgRNAs-targeting *HPRT1* TSS effectively rescued 6TG-induced proliferation arrest in Cas9-expressing cells (Fig. 1b and Supplementary Fig. 1b), suggesting that sgRNAs-targeting TSS also directed effective Cas9-mediated gene suppression. Consistent with these observations, sgRNAs targeting the TSS of core essential genes[12,13] induced a proliferation arrest in cells expressing Cas9 or KRAB-dCas9 (Fig. 1c,d). Moreover, sgRNAs-targeting oncogenes, including *KRAS* and *BRAF*, induced a proliferation arrest in cell lines harbouring mutations in these oncogenes upon Cas9 or

KRAB-dCas9 expression (Fig. 1e,f and Supplementary Fig. 1f). These observations demonstrate that KRAB-dCas9 can be used to identify cell essential genes. However, in contrast to what we observed with CRISPRc, only a limited number of sgRNAs (40% of TSS sgRNAs) were effective in CRISRPi (Supplementary Fig. 1c).

**Prediction of effective sgRNAs in KRAB-dCas9 screens.** To refine the selection of sgRNAs in library design, we developed an algorithm that predicts effective sgRNAs in KRAB-dCas9 experiments. Prior reports have demonstrated that nucleosome positioning dictates access of sgRNA to DNA[14]. We hypothesized that the distance between the target locus and the TSS position would predict whether a sgRNA accesses DNA. To test this hypothesis, we trained a support vector machine (SVM) using previously published CRISPRi data[14] to model the sgRNA efficiency as a non-linear function of target-TSS distance. We found that sgRNAs targeting 100 bp downstream of the TSS induced the most effective proliferation arrest phenotype (Supplementary Fig. 1d). Furthermore, in consonance with previous reports[15], the SVM model showed higher predictive power when using CAGE-seq (FANTOM) data for TSS annotation (Supplementary Fig. 1e,f), in comparison to NCBI RefSeq annotations. Using these considerations, we found that the SVM model improved sgRNA selection by increasing the fraction of effective sgRNAs from 40 to 70% (Supplementary Fig. 1g,h).

**Impact of alternative TSSs on *KRAB-dCas9* gene inhibition.** Among 34 genes in the tiling sgRNA experiment, we found that 13 exhibited discordant viability scores in Cas9 and KRAB-dCas9-expressing cells (Supplementary Fig. 1c), including the oncogene *MYC*. Further examination of publicly available CAGE-seq data[16] showed that *MYC* is expressed from two alternative TSSs, whereas the tiling sgRNA library only targeted the secondary TSS that is annotated in the NCBI RefSeq database. This finding implicates alternative TSSs as a potential explanation for the differential viability scores observed.

To explore the impact of alternative TSSs on KRAB-dCas9 experiments, we constructed a lentivirally delivered sgRNA library targeting 264 genes, including 104 core essential genes, 105 cancer-related genes, 55 amplified genes (in two different amplicons) and 1,064 negative control sgRNAs (sgRNAs that do not map to the human genome, or target the AASV locus). For each gene, up to three alternative TSS clusters defined in CAGE-seq annotation were targeted (Supplementary Data 3). Coding exons were also targeted for comparison (6 sgRNAs per gene). Following introduction of this library to HT29 cells expressing Cas9 or KRAB-dCas9, we used massively parallel sequencing to measure sgRNA abundance. We found that sgRNAs targeting the primary TSS (highest CAGE-seq peak) were most effective in inducing proliferation arrest in cells expressing KRAB-dCas9 (Fig. 2a). For example, only sgRNAs targeting the primary TSS of *MYC* but not the secondary TSS had an expected proliferation effect (Fig. 2b). Using sgRNAs-targeting primary TSSs, 96 of 104 (92%) previously defined core essential genes[12] were detectable ($P < 0.01$, $Z$-test) in KRAB-dCas9 experiment (Supplementary Fig. 2a,b). These observations suggest rational selection of TSS based on CAGE-seq improves KRAB-dCas9 library design, by which KRAB-dCas9 knockdown can achieve high sensitivity close to Cas9 knockout (98%) (Supplementary Fig. 2a,b).

**KRAB-dCas9 is not affected by cleavage-induced arrest.** We and others have previously reported a gene-independent clea-vage-induced proliferation arrest in Cas9-mediated CRISPRc

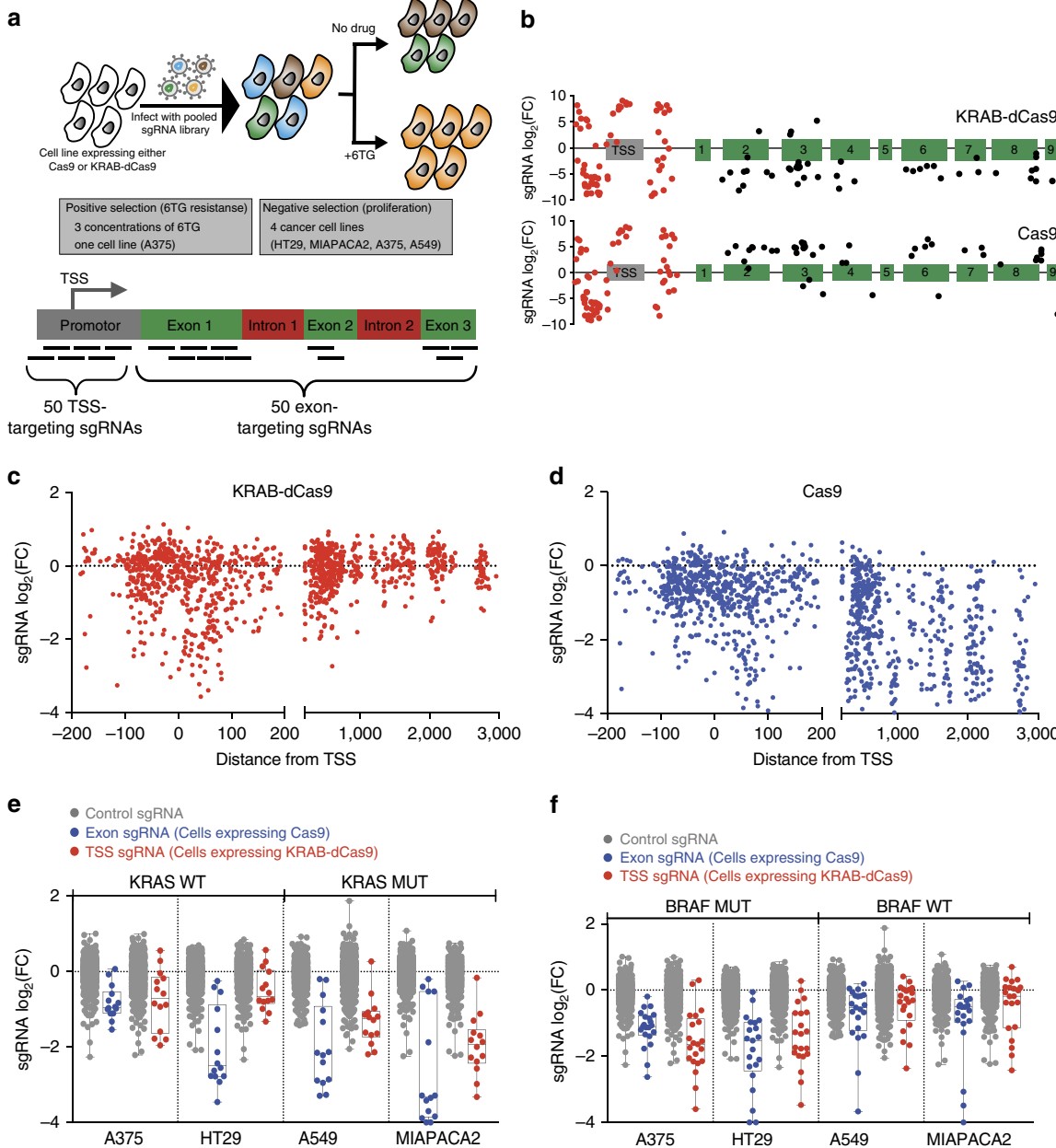

**Figure 1 | Loss-of-function screens performed in cells expressing Cas9 and KRAB-dCas9 using tiling sgRNA library. (a)** Strategy used for design and assessment of the tiling sgRNA library. **(b)** Proliferation changes fold-change (FC) induced by sgRNAs-targeting *HPRT1* following treatment of A375 cells with 6TG (15 μM) in cells expressing either Cas9 or KRAB-dCas9. **(c)** Proliferation changes induced by sgRNAs targeting seven cell essential genes (*POLR1C, POLR2D, RPL19, RPL4, RPL5, RPL8, U2AF1*) in KRAB-dCas9-expressing cells. Each point represents the mean sgRNA score across four cell lines. **(d)** Cells expressing Cas9 and the sgRNAs from **c**. **(e)** Proliferation changes induced by control or *KRAS*-targeting sgRNAs in cells expressing KRAB-dCas9 (red) or Cas9 (blue). Only efficient sgRNAs (as determined by the sgRNA predictive algorithm) are shown for KRAB-dCas9 and only exon-targeting sgRNAs are shown for Cas9. **(f)** Proliferation changes induced by control or *BRAF*-targeting sgRNAs. Error bars represent the deviation from the mean of sgRNAs targeting the indicated genes. All FCs are presented as the mean of duplicate experiments.

screens[5,6]. To evaluate whether such effects are also induced by KRAB-dCas9, we measured the proliferation effects induced by sgRNAs targeting a genomic locus that does not contain coding genes (AAVS1). In contrast to endonuclease cleavage-induced proliferation arrest in Cas9 expressing cells (Fig. 2c), we did not detect an effect of KRAB-dCas9 on proliferation (Fig. 2d). Furthermore, sgRNA spacers aligned to multiple genomic regions induced a strong anti-proliferative effect in Cas9, but not in KRAB-dCas9-expressing cells (Fig. 2e). We further expanded this analysis by allowing imperfect matches to the human genome (1 mismatch or only part of the spacer was used for alignment)

and found that sgRNAs aligned to multiple genomic loci directing KRAB-dCas9 exhibited no effect on proliferation (Supplementary Fig. 2c–e). These observations suggest that gene silencing by KRAB-dCas9 is not prone to a proliferation arrest effect induced by Cas9 endonuclease activity.

We next evaluated the consequence of using Cas9- or KRAB-dCas9 to suppress genes located in two highly amplified regions—8q24 (containing *MYC*) and 17q12 (containing *HNF1B*)—in HT29 cells (Fig. 2f,g). As expected, we observed a clear proliferative defect following Cas9-mediated deletion of all genes in these amplicons. In contrast, KRAB-dCas9-mediated

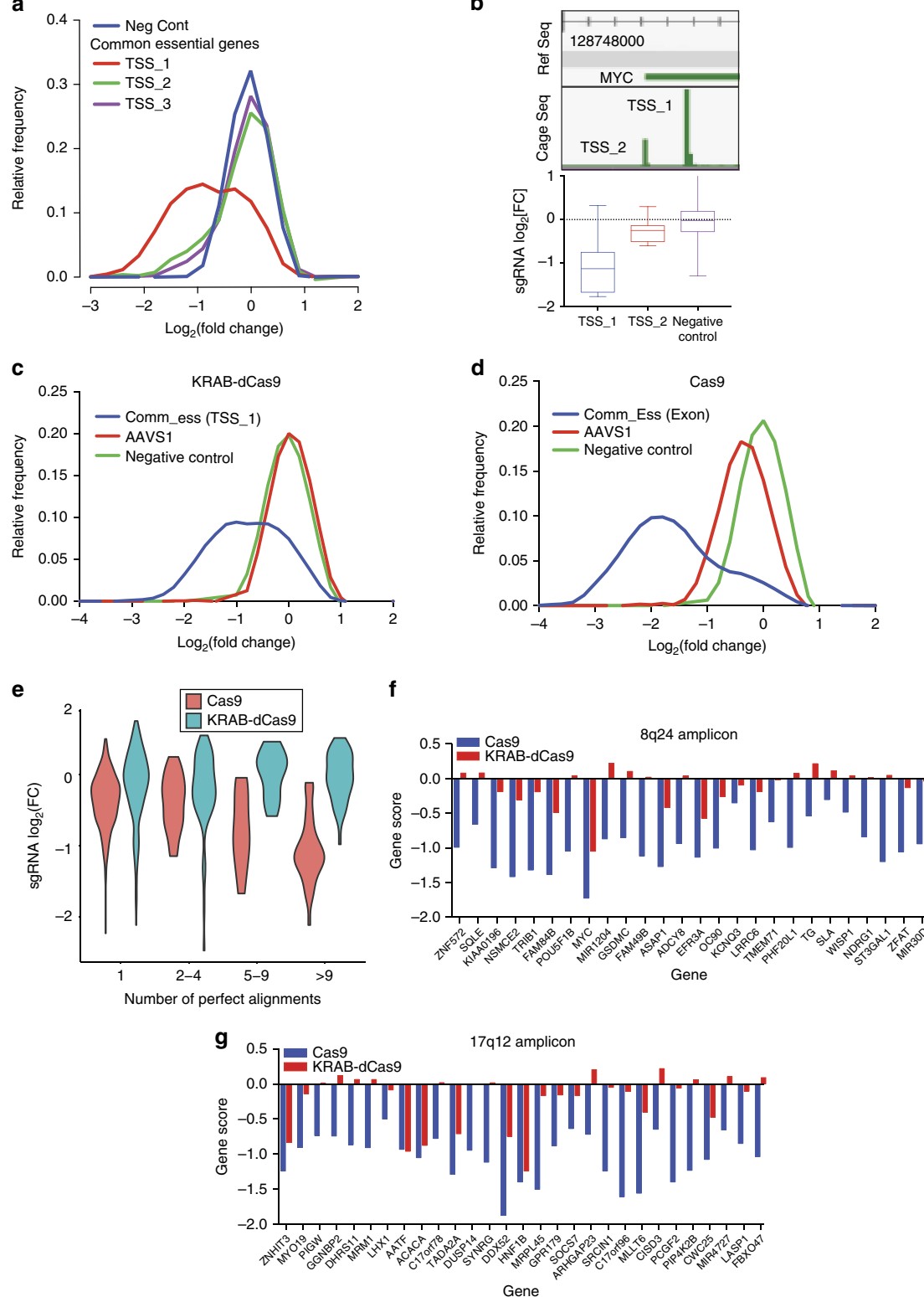

**Figure 2 | Proliferation changes induced by KRAB-dCas9 are not prone to cleavage-induced toxicity.** (**a**) Distribution of proliferation changes induced by sgRNAs targeting the three top cage-seq TSS of 105 cell essential genes in HT29 cells expressing KRAB-dcas9. (**b**) Proliferation changes and locations of two alternative *MYC* TSSs (TSS_1 and TSS_2) in HT29 cells expressing KRAB-dCas9. Error bars represent the deviation from the mean of sgRNAs targeting the indicated gene. (**c**) Distribution of proliferation changes induced by sgRNAs targeting the AASV1 genomic region in HT29 cells expressing Cas9 (**d**) Distribution of proliferation changes induced by the same AAVS1-targeting sgRNAs from **c** in cells expressing KRAB-dCAs9. (**e**) Violin plot showing distribution of proliferation changes induced by sgRNAs that target multiple genomic regions in Cas9 (red) or KRAB-dCAs9 (blue) expressing cells. (**f**) Proliferation changes induced by Cas9 (blue) or KRAB-dCas9 (red) following suppression of genes within the 8q24 amplicon in HT29 cells. Gene score were calculated by the mean proliferation changes induced by all sgRNAs-targeting TSS_1 (KRAB-dCas9) or exon (Cas9). (**g**) Experiment as (**f**) but targeting the 17q12 amplicon. All FCs are presented as the mean of duplicate experiments.

suppression of genes on the 8q24 amplicon identified the oncogene *MYC* as the only dependency in this amplicon (Fig. 2f). Furthermore, in agreement with our previous observations using RNAi[17] KRAB-dCas9-mediated suppression of 17q12 genes identified *HNF1B* as the gene required for proliferation in this region (Fig. 2g). These observations demonstrate that gene suppression using Cas9 and KRAB-dCas9 provides complementary information.

**Bidirectional promoters bias CRISPRi screens**. To evaluate Cas9 and KRAB-dCas9 in a genome scale format, we constructed an sgRNA library targeting 19,818 protein-coding regions and 1,812 non-coding RNAs (Supplementary Data 4). We designed five sgRNAs for each CAGE-seq primary TSS, introduced this library into HT29 cells expressing KRAB-dCas9, and compared the results to a CRISPR-Cas9 screen performed in the same cells[5].

Using a quantitative approach based on false discovery rate (FDR) thresholds (Fig. 3a and Supplementary Data 5), we identified 180 genes to be essential in both approaches with high confidence. In addition, we found 175 genes that scored only in cells expressing wild-type Cas9, and 57 genes that scored only in cells expressing KRAB-Cas9.

As expected, we found that the Cas9-specific category is enriched with amplified genes (Fig. 3b and Supplementary Data 6), which confirmed that cleavage based toxicity causes false positives in Cas9 screens. In contrast, we found that the genes that scored only in cells expressing KRAB-dCas9 were enriched for genes transcribed from a bidirectional promoter (Fig. 3c and Supplementary Data 6). For example, *TRMT5* and *SLC38A* are located on chromosome 14 and are expressed from a common promoter but in opposing orientations (Fig. 3d). We found a similar pattern and magnitude in both directions in HT29 cells when we performed chromatin immunoprecipitation (ChIP)-seq

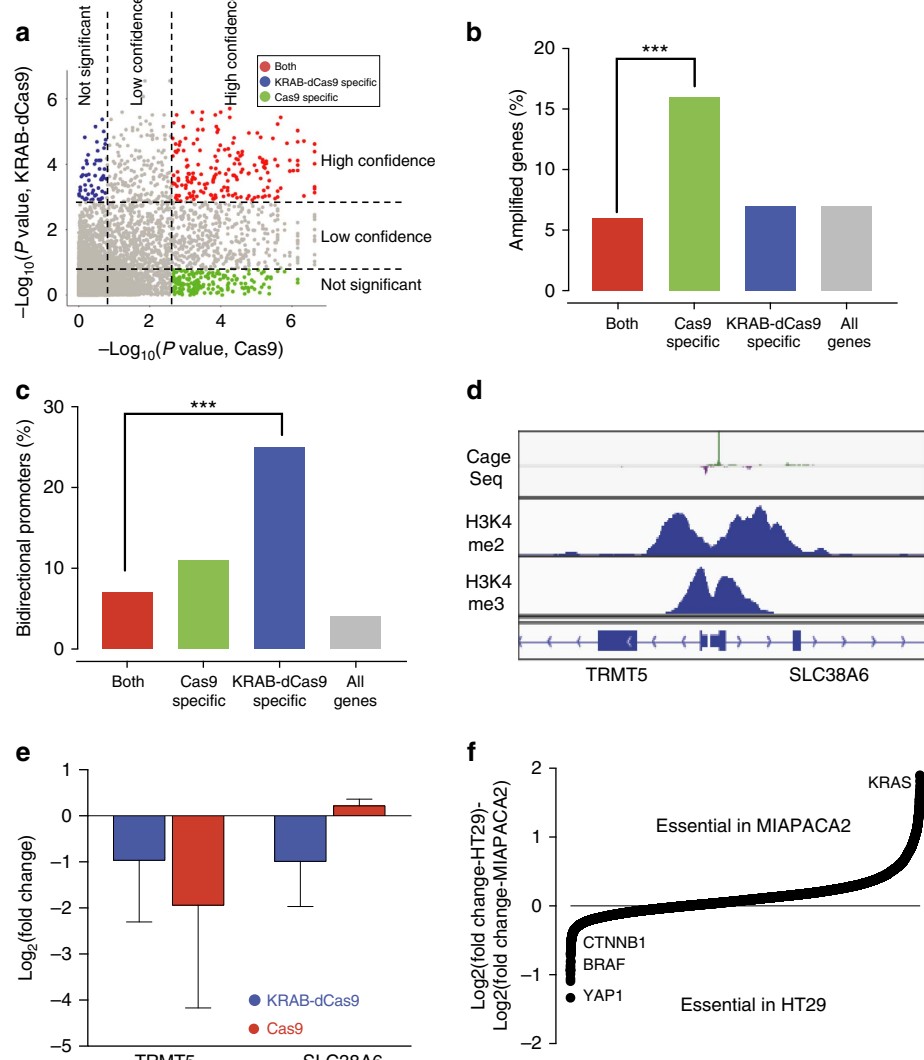

**Figure 3 | Genome scale CRISPRi proliferation screens.** (**a**) MAGECK[22] was used to identify statistically significant cell essential genes in HT29 cells expressing Cas9 or KRAB-dCas9. Genes scored to be essential were categorized into three groups: 'both' (essential in both Cas9 and KRAB-dCas9), 'KRAB-dCas9 specific' and 'Cas9 specific'. (**b**) The percentage (%) of amplified genes in three categories. ***$P < 0.001$, Fisher's exact test. (**c**) The percentage (%) of genes containing bidirectional promoters in three categories. ***$P < 0.001$, Fisher's exact test. (**d**) Peaks of cage-seq (from ref. 16) and ChIP-seq peaks of H3K4-me2 and H3K4-me3 in the promoter region of *TRMT5* and *SLC38A6*. (**e**) Proliferation changes induced by KRAB-dCas9 (blue) or Cas9 (red) in HT29 cells. Error bars represent deviation from the mean of all sgRNAs targeting the indicated genes. (**f**) Comparison of genetic dependencies between HT29 and MIAPACA2 cells expressing KRAB-dCas9. Genes are ranked based on the Log2[FC] difference between MIAPACA2 and HT29 cells. All FCs are presented as the mean of duplicate experiments.

profiling of the active transcription initiation markers H3K4-me2 and H3K4-me3 (Fig. 3d). We found that both genes scored as essential in cells expressing KRAB-dCas9, but only *TRMT5* scored in cells expressing wild-type Cas9 (Fig. 3e), suggesting *SLC38A* as an off-target in KRAB-dCas9 due to close proximity of their TSSs.

We further evaluated the extent of the bidirectional promoter effect by measuring the number of genes that are considered as cell essential in CRISPRi as a function of their distance from a core cell essential gene (Supplementary Fig. 2e). We found that genes that are located up to 1 kb from a cell essential gene are more likely to be considered as cell essential. On the basis of this observation, we re-evaluated CAGE-seq promoter annotations and found that at least 12.9% of protein-coding genes share bidirectional promoters within a distance of 1 kb (Supplementary Fig. 2f), which introduces potential false positives in KRAB-dCas9 experiments. In addition, we used this library to compare cell essential genes in HT29 and a KRAS-mutant pancreatic cancer line MIAPACA2 (Fig. 3f). Confirming prior reports[4], we found oncogenes *CTNNB1*, *BRAF* and *YAP1* as essential for the proliferation of HT29, and *KRAS* to be essential for the proliferation of MIAPACA2. These observations demonstrate that KRAB-dCas9 is an alternative approach for identifying cell-specific or condition-specific essential genes.

## Discussion

Here we confirm that both CRISPRc and CRISPRi can be used to identify essential genes in mammalian cells. We found that compared to CRISPRc, identifying effective sgRNAs in CRISPRi requires information about the structure of the 5′ untranslated region upstream of target genes. These observations confirm and extend a recent report that demonstrated that nucleosome positioning predicts CRISPRi sgRNA efficiency[14]. Here, we found that CAGE-seq is a useful approach to identify active TSS and facilitates the design of sgRNA libraries for the use in CRISRPi experiments. We anticipate that the iterative use of CAGE-seq and other annotations of untranslated regions with focused CRISPRi experiments will lead to a better definition of transcription start sites and improved sgRNA libraries (summarized in Table 1).

By comparing CRISPRc and CRISPRi in the same cell lines, we found that both of these approaches allow one to identify cell essential genes; however, the two different approaches exhibit different limitations. Specifically, similar to previous reports[5,6], we observed that CRISPRc targeting of amplified genomic regions showed a gene-independent proliferation effect, and CRISPRi targeting of bidirectional promoters can be a source of false positive results. We found that bidirectional promoters may be present in up to 10% of human genes and therefore affects the interpretation of CRISPRi loss-of-function screens.

Our preliminary observations suggest that understanding the effects of targeting KRAB-dCas9 to specific sites near bidirectional promoters requires the interrogation of many sgRNA and further work is required to define the features of TSS sgRNAs that allow one to avoid effects on bidirectional promoters. On the basis of our observations, we suggest that complementary screening strategies that combine these two gene suppression approaches will recover highly specific cell essential genes.

## Methods

**Pooled sgRNA libraries.** Pooled sgRNA libraries were prepared by ligating a pooled PCR amplified oligopool (CustomArray) into the BsmBI sites of pXRP003 (addgene # 52963)[3,4]. Supplementary Data 2–4 contain the sequences of individual sgRNAs in each library. The following principles were used for design of each of the three libraries.

Pooled sgRNA library 1—Focused tiling sgRNA library. The tiling library includes sgRNAs targeting 7 core essential genes, 21 oncogenes, 5 tumour suppressor genes and *HPRT1* (Supplementary Data 1 and 2). For each gene, we designed ∼50 sgRNAs targeting the coding exons (downstream of the first exon). In addition, for each of these genes we designed ∼100 sgRNAs targeting the RefSeq TSS locus (starting from 100 bps upstream of RefSeq TSS). On average, for each gene ∼50 sgRNAs target the TSS region (±100 bps relative to the TSS), and the rest target intronic regions. The only criteria used for selection of these sgRNAs was the 'NGG' motif. In addition, we included 979 non-specific sgRNAs (not mappable to the human genome) as negative controls.

Pooled sgRNA library 2—Multiple-TSS sgRNA library. The multiple-TSS library includes sgRNAs-targeting promoters and exons of 104 core essential genes, 105 cancer-related genes and 55 genes in amplified regions (Supplementary Data 3). To define a list of core essential genes, we used 217 constitutive core essential genes identified from RNAi datasets[12]. Among them, we selected 104 genes based on published CRISPR-Cas9 screens based on the following criteria: (i) genes shown to be essential in both HL-60 and KBM-7 cells[18] for which sgRNA the available in the Wang et al.[19] library and (ii) genes that have homologues in mouse embryonic stem cells that were also shown to be essential[20]. For each gene, up to three promoters were selected based on FANTOM5 (ref. 16) annotation (three highest CAGE-seq peaks) using the following criteria: (1) 20 nt upstream of an NGG motif within ±200 bps from CAGE-seq peak; (2) Top seven ranking sgRNA using the SVM model (see below). (3) sgRNA sequences with perfect alignment to multiple genomic loci (using Bowtie v1) were excluded; (4) Sequences with four consecutive T's were excluded; (5) sgRNA sequences with at least 80% C or G were excluded. Exon-targeting sgRNAs were extracted from GeCKO v2 library[21]. In addition, we included 714 negative control sgRNAs (non mappable to the human genome) and 267 sgRNAs-targeting AAVS1 locus as negative controls.

Pooled sgRNA library 3-Genome-scale KRAB-dCas9 library. We overlapped RefSeq gene list with FANTOM5 promoter annotation by gene symbol and identified 17,552 unique genes for genome-scale CRISPRi library (Supplementary Data 4). For each gene, we designed 5 sgRNAs targeting the primary TSS (using the same criteria described above for library 2). In addition, 714 non-specific and 267 AAVS1-targeting sgRNAs were included as negative controls.

**Proliferation screens.** Each proliferation screen was performed in duplicate. Virus particles containing the above described pooled sgRNA libraries were used to infect the specified cell lines at a multiplicity of infection (MOI) of 30% (500–1,000 cells per sgRNA). Following transduction, infected cells were selected using Puromycin (2 µg ml$^{-1}$) for 4 days. In 6TG rescue screens 6TG (Sigma, USA) at the indicated concentration was added after 7 days in culture. Genomic DNA extracted following 21 days in culture was used for Illumina sequencing[3,4].

| Table 1 | Pros and cons of CRISPRc and CRISPRi. | | |
|---|---|---|
| **Feature** | **CRISPRc** | **CRISPRi** |
| Average strength of observed phenotype | Strong | Medium |
| sgRNA design | Target sequence drives selection. Easy to identify efficient sgRNAs | Only a small number of sgRNAs give a desired phenotype. Identification of TSS necessary. |
| Tissue specificity | No specific concern | Differential promoter usage in specific tissues or context should be considered |
| False positives | Amplified genomic regions | Bidirectional promoters |
| False negatives | Exon skipping | Genes expressed from multiple promoters |

**Data processing.** The sgRNA abundance were measured as read counts in each experiment, followed by log transformation:

$$y_{ij} = \log\left(\frac{x_{ij}}{c_j} + \varepsilon\right)$$

where $x_{ij}$ is the read counts of the $i$th sgRNA in $j$th experiment. $c_j$ is a normalization factor in $j$th experiment, which is the trimmed mean (10–90 percentile) of read counts for all non-specific sgRNAs. We used trimmed mean to exclude outliers that are often observed in positive selection experiments. $\varepsilon$ is a small constant for stabilize variations when read counts is small, and is set to be 0.05. We merged biological replicates by averaging log-transformed sgRNA abundance. Relative sgRNA abundance (log fold-change) was computed by comparing sgRNA abundance before and after treatment in log-transformed space.

**Prediction of sgRNA efficiency in CRISPRi screens.** We modelled sgRNA efficiency as a non-linear function of the distance between its target DNA locus and the TSS of the target gene in a CRISPR/dCas9 experiment, that is,

$$S_x = f(d_x)$$

where $x$ is an sgRNA, $S_x$ is the efficiency score of $x$, and $d_x$ is a signed distance between the 3′ end of DNA targeted by $x$ (position of 'N' in 'NGG' PAM motif) and the TSS of its target gene. $d_x < 0$ for $x$ targeting upstream of TSS, and $d_x > 0$ for $x$ targeting downstream of TSS.

We collected 18,380 sgRNAs targeting 1,539 genes in Horlbeck et al.[14]'s study, where each sgRNA was assigned an activation score representing sgRNA efficiency in original paper. To refine the dataset for model fitting, we selected the genes that: (i) overlap with RefSeq and FANTOM gene annotation by gene symbol; (ii) with at least seven sgRNAs; and (iii) the difference of 25 and 75 percentile of activation scores larger than 0.5. By these, we selected 576 genes that have sufficient numbers of efficient and inefficient sgRNA for model fitting, corresponding to 5,548 sgRNAs in total.

To model the non-linear relationship between $S_x$ and $d_x$, we applied a kernel-based SVM regression method ($\varepsilon$-regression), by which a weight corresponding to predicted efficiency was computed for each $d_x$. We trained two SVMs for prediction, corresponding to RefSeq TSS annotation (https://genome.ucsc.edu/, hg19 assembly) and FANTOM5 TSS annotation (http://fantom.gsc.riken.jp/, robust promoters). For FANTOM annotation, the center of the primary promoter cluster (P1) was used as TSS locus in computation. The trained models are shown in Supplementary Fig. 1c,d. We compared the predictive powers of two models by cross-validation, where two-third of the genes were randomly selected for training, and the rest one-third were used for testing (Supplementary Fig. 1e).

**Identification of essential genes in pooled sgRNA library 2.** Since Cas9 screens are subject to DNA-cleavage toxicity, we used AAVS1-targeting sgRNAs as negative controls for calling essential genes in Cas9 and KRAB-dCas9 screens with multi-TSS library. We used Z-test to determine the statistical significance of gene essentiality, where Z-score is defined as follow:

$$Z(g) = \frac{m_g - m_{AAVS}}{\sigma_{AAVS}} \times \sqrt{N_g}$$

where g is a gene, $m_g$ is the average log fold-change of sgRNAs targeting g, $m_{AAVS}$ and $\sigma_{AAVS}$ are the mean and standard deviation of sgRNAs-targeting AAVS1 locus, and $N_g$ is the number of sgRNAs targeting g. For Cas9 screens, sgRNAs are those targeting exon of g; for KRAB-dCas9 screens, sgRNAs are those targeting primary TSS of g.

**Identification of essential genes in pooled sgRNA library 3.** We used the MAGeCK pipeline[22] (https://sourceforge.net/projects/mageck/) to call essential genes in genome-wide Cas9 or KRAB-dCas9 screens. Default settings of MAGeCK were used in the analysis.

**Cell lines.** Cell lines were obtained from the Broad institute Biological Samples platform and regularly tested for mycoplasma contaminations. HT29, MIAPACA2 and A549 were cultured in DMEM with 10% of fetal bovine serum (Sigma) and A375 was cultured in RPMI1650 (Sigma) containing 10% fetal bovine serum (Sigma).

**Chromatin immunoprecipitation.** Each chromatin immunoprecipitation experiment was done in triplicate. Cells grown on a monolayer were fixed for 15 min at room temperature with 1% formaldehyde. The crosslinking reaction was stopped by adding 2.5 M glycine and 10 min at room temperature. Cells were washed twice with cold PBS and then collected by scraping directly into RIPA buffer (150 mM NaCl, 50 mM Tris pH 8, 5 mM EDTA, 1% IGEPAL, 0.5% sodium deoxycholate, 0.1% SDS, protease inhibitors) on ice. Cells were then sonicated, and the lysates obtained were centrifuged at 9,300 g 4 °C for 10 min. The supernatant was quantified using a BCA protein assay kit (Pierce) and diluted to 1 mg ml$^{-1}$ with RIPA buffer.

For immunoprecipitation experiments, 1 mg of cell lysate was pre-cleared by incubation with RIPA-washed Protein-G sepharose beads (GE Healthcare) at 4 °C for 1 h. Pre-cleared supernatant was incubated with antibody against H3K4Me3 (Milipore, CS200580) overnight at 4 °C. The next day, 50 μl of Protein-G sepharose beads were added for 2 h at 4 °C. The beads were then washed twice with cold RIPA following by 4 washes with wash buffer (100 mM Tris-HCl pH = 8.5, 500 mM LiCl, 1% NP-40 (v/v), 1% deoxycholic acid (v/v)). Beads were then washed again twice with RIPA buffer and then incubated with 50 μl of TE buffer. The DNA was then reverse cross-linked by adding 200 μl of Talianidis buffer (70 mM Tris-HCl pH = 8, 1 mM EDTA, 1.5% SDS (w/v)) and incubating for 10 min at 65 °C. The beads were then centrifuged and the supernatant containing DNA was collected. To reverse cross-link, NaCl was added to a final concentration of 200 mM, followed by incubation at 65 °C for 6 h. To digest the remaining protein, 20 μg proteinase K was added and incubated at 45 °C for 30 min. Immunoprecipitated DNA was then purified by phenol/chloroform extraction, ethanol precipitated, then by MinElute Reaction Cleanup column (Qiagen). In parallel, input DNA was prepared using cell lysate without the immunoprecipitation steps. ChIP DNA was quantified by Quant-iT dsDNA HS Assay (Invitrogen).

Overall, 20 ng of ChIP DNA or whole-cell extract were used to generate an Illumina sequencing library. DNA fragments were end-repaired using the End-It DNA End-Repair Kit (Epicentre) and then a single 'A' base was added using Klenow (NEB). The fragments were ligated with Illumina Indexed adaptors (TruSeq DNA Sample Prep Kits) using DNA ligase (NEB). The ligated product was selected for 300–400 bp on 2% agarose gel to remove the non-ligated adaptors and was subjected to 18 PCR cycles with Illumina PCR primer cocktail (TruSeq DNA Sample Prep Kits). PCR products were purified on 2% agarose gel to retain fragment between 300 and 400 bp. Library concentrations were quantified by Qubit fluorometer (Invitrogen) and by quantitative PCR (Kapa Biosystem). Two barcoded libraries were pooled and sequenced to 50 bp in a single lane on Illumina HiSeq2000. The MACS analysis method[23] was used for peak calling.

**Data availability.** All CRISPR screening datasets generated in this study are available as raw read counts (Supplementary Data 7–10). All other data are available from the authors upon reasonable request.

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

## Acknowledgements

We wish to thank Dr Andrew Aguirre and Dr John Doench as well as members of the Hahn laboratory for helpful discussions. We thank Tamara Mason for sequencing these samples. This work was supported by: The Svenson Family foundation (J.R.), NIH U01 CA176058 (W.C.H.) and NIH U01 CA199253 (W.C.H.).

## Author contributions

J.R., H.X. and W.C.H. designed and performed the experiments. J.R., W.H. and F.V. performed screens, S.G. performed validation experiments, X.W. performed ChIP-Seq experiments, J.R., H.X., F.V., D.E.R., A.T. performed data analysis. J.R., H.X., W.C.H. wrote the manuscript. W.C.H. supervised the studies.

## Additional information

**Competing interests:** The authors declare no competing financial interests.

