## [Peer Review File · Nature Communications]

Reviewers' Comments:

Reviewer #1 (Remarks to the Author)

In this study, Rosenbluh et al reported direct comparison of two screening approaches: one is genome editing-based method using CRISPR-Cas9 (cutting, CRISPRc) and the other is gene repression-based method using KRAB-dCas9 (transcription, CRISPRi). The authors used two parallel loss-of-function screens to identify essential proliferation and/or survival genes in a few cancer cell lines. Following up their prior work (Aguirre et al, Cancer Discovery 2016) they demonstrate here that CRISPRc may generate systemic off-target effect in highly amplified genomic regions, and showed CRISPRi can provide additional information to accurately identify genes that are critical for the growth phenotype located in such regions. Conversely, CRISPRi screens can generate false-positive hits for genes regulated by bidirectional promoters; and CRISPRc can be used to distinguish this off-target effect. Therefore, CRISPRc and CRISPRi screens can yield complementary information in identifying key candidate driver genes for oncology research. The results reported are of general interest to the CRISPR community, but the manuscript needs further improvements before considering for acceptance.

1. All experiments were performed on the large-scale in a pooled manner. The authors relied on previous knowledge that "a gene is an essential gene". To validate this claim and confirm "the gene is indeed an essential gene", they should validate some of their sgRNA hits for CRISPRc and CRISPRi in a non-pooled manner.
2. The bi-directional off-target results are interesting. The authors should propose how many of these genes in the genome. Since the human genome is sparse with only 2% coding proteins, is this a major limitation for CRISPRi? Also, is there a way to design sgRNAs to avoid such off-target effects? The authors suggested targeting within 100bp downstream of TSS to be most effective. Following this rule, is it possible to avoid bi-directional off-target effects?
3. There are several conclusions comparing CRISPRc and CRISPRi, with each having some pros and cons. For readers to better understand these points, this reviewer suggests the authors to include a Table in the manuscript that summarizes and compares the features of CRISPRc and CRISPRi.
4. The authors demonstrated that alternative TSSs can impact KRAB-dCas9 gene inhibition. Based on CAGE-seq and H3-K4-me3 Seq data in HT29 cells, it would be nice to demonstrate targeting TSS1 and TSS2 PELO and determine if it leads to effective CRISPRi suppression (Supplementary Figure 3B).
5. In page 7, the authors indicated that they re-evaluated CAGE-seq promoter annotations and found 4.7% of protein-coding genes share bidirectional promoters (Fig. S3D). However, this figure is missing in the supplementary information.
6. There are a number of typos in the manuscript and mistakes in the figures, and please carefully revise the manuscript and correct them. For example, Supplementary Figure legends 3A and 3B are reversed; Fig 1B might have wrong labels for "Cas9" and "KRAB-dCas9".
7. The references: referring to the CRISPRi method should include original publications on the topic as reported in Qi et al. 2013; Mali et al. 2013; etc.

Reviewer #2 (Remarks to the Author)

In this study, Rosenbluh et al demonstrated that CRISPRi screens can provide complementary phenotypic information to CRISPRc (CRISPR-cutting) screens. The authors show that both CRISPRi (after optimization of library design) and CRISPRc screens can recover most known essential genes, indicating low rates of false negatives. However, they found that both CRISPRc and CRISPRi have significant numbers of false positives. Importantly, they show that the source of false-positives with these two loss-of-function screening modalities is generated by distinct (non-overlapping) mechanisms; copy number effects for CRISPRc and bidirectional promoters for

CRISPRi. Therefore, both types of screens serve to address the shortcomings of the other, and when combined can provide a more complete and accurate picture of gene function. The findings of this study significantly increase our understanding of the types of artifacts that can arise with CRISPRi compared to CRISPRc, and therefore will be of strong interest to the broader research community. The concerns outlined below should be addressed before publication:

1. The authors demonstrated that TSS-targeting guides exhibited very similar phenotypes regardless of whether Cas9 or dCas9-KRAB were used for many genes, even for untranslated regions (Figs 1B, 1D, S1B). This is quite a surprising finding, given that the principal mechanism of Cas9 gene inactivation is thought to be by inducing frameshift mutations in protein coding regions. Upon further inspection, I noticed that many of the TSS-targeting guides exhibiting the strongest lethal phenotypes with Cas9 contain low complexity repeats (eg. GCCGCCGCCGCCGCC). This is likely due to how the tiling library guides were chosen with no filtering for multi-cutters or low complexity sequences. The authors should address whether these TSS regions are inherently less complex and therefore more likely to cause non-specific lethality due to multi-site cutting (as described in the prior work cited by the authors). I would strongly recommend to either remove (or at least clearly label in the graphs) all potential multi-cutters and low complexity sequences before performing downstream analysis, as it may be biasing results.

2. Is there any evidence of these same low-complexity sequences showing increased lethality with dCas9-KRAB by promiscuous multi-targeting of the TSS (or other genomic regions) for many transcripts? Fig 2E addresses perfect matches, but not potential repeats that don't match completely to other sites in the genome (a very common 17-mer or similar).

3. Raw count data should be provided in the supplementary data files so that re-normalization can be performed. Additionally, the annotations in Fig1C,D (distance from TSS) are not annotated in the supplementary data, and should be added.

4. Minor point : The finding that CRISPRi screens targeting bidirectional promoters can lead to false-positives is an important finding. I wonder if it is at all possible to identify sgRNAs around bi-directional promoters that preferentially silence one transcript over the other, or if always both genes are strongly impacted (which may be the case due to epigenetic silencing spread by KRAB domain). One way to address this question is to run a RT-PCR for both genes for a number of the sgRNAs in the tiling array? Or can the authors think of a way to addressing this question with their current dataset?

Detailed responses to the reviewers

Reviewer #1:

1. All experiments were performed on the large-scale in a pooled manner. The authors relied on previous knowledge that “a gene is an essential gene”. To validate this claim and confirm “the gene is indeed an essential gene”, they should validate some of their sgRNA hits for CRISPRc and CRISPRi in a non-pooled manner.

Response:

Over the past several years, several groups have identified genes that are essential in human cells (Wang et al. *Science*, 2015, Blomen et al. *Science*, 2015). For this manuscript, we used a gene set that has been previously validated in several different tissue types (Hart et al. *Cell*, 2015) and has been used by others as a positive control set for optimization of genetic screens (Doench et al. *Nature Biotechnology*, 2016). Specifically, we selected 104 genes based on published CRISPR-Cas9 screens based on the following criteria: (i) genes shown to be essential in both HL-60 and KBM-7 cells (Wang et al. *Science*, 2014) for which sgRNA are available and (ii) genes that have homologs in mouse embryonic stem cells that were also shown to be essential (Koike-Yusa et al. *Nature Biotechnology*, 2014).

Similar to what others have observed, we found that suppression or deletion of genes in this essential gene set inhibits proliferation (Fig. 2A, C and D). However, we agree with the reviewer that we did not describe how we defined these essential genes in the original manuscript. We have added a more detailed description of how we selected core essential genes (Supplementary methods *Pooled sgRNA library 2*) as well as specific citations of these reports (Hart et al. and Doench et al.) to the revised manuscript.

2. The bi-directional off-target results are interesting. The authors should propose how many of these genes in the genome. Since the human genome is sparse with only 2% coding proteins, is this a major limitation for CRISPRi? Also, is there a way to design sgRNAs to avoid such off-target effects? The authors suggested targeting within 100bp downstream of TSS to be most effective. Following this rule, is it possible to avoid bi-directional off-target effects?

Response:

The reviewer asked two questions.

- a) What is the frequency of bidirectional promoters? Although protein coding genes occupy only 2% of the human genome, such genes are not uniformly distributed but instead are clustered, resulting in an increased occurrence of bidirectional promoters over what might be expected if protein coding genes were randomly distributed. In the original manuscript, we used a conservative cutoff for defining bidirectional promoters (less than 100 bp apart) and found that 4.7% of RefSeq genes contain a bidirectional promoter. To further address the reviewer’s question, we have now extended our analysis by calculating the percentage of genes that are located near a core essential gene and are identified as cell essential in CRISPRi (Fig. S2F). This new analysis, which likely underestimates the occurrence of bidirectional promoters, demonstrates that targeting promoters that are located within 1000 bp from each other can be affected by this bidirectional promoter effect. Based on this new analysis, we reanalyzed the prevalence of bidirectional promoters in the human genome and found that 7.13% of RefSeq genes are within 200 bp from a TSS of another gene transcribed in a different direction. When we increased the threshold to 1000 bp, 12.95% of RefSeq genes are associated with bi-directional promoters. We have added a new figure to the revised manuscript that describes these findings (Fig. S2F).

- b) Can one design libraries to avoid bidirectional promoters? Our analysis of bidirectional promoters relied on information derived using a genome scale CRISPRi library, which targets sequences within the first 100 bps of the TSS. Furthermore, our new analyses show that this effect is observed even if promoters are found within 1000 bp (Fig. S2G). It is clear that further work will be necessary to develop libraries that avoid these issues. In the short term, suggest the use of combined CRISPRi and CRISPRc as an approach to identify this effect. We have added text to the revised manuscript to discuss these issues (p. 8 in the revised manuscript).
3. There are several conclusions comparing CRISPRc and CRISPRi, with each having some pros and cons. For readers to better understand these points, this reviewer suggests the authors to include a Table in the manuscript that summarizes and compares the features of CRISPRc and CRISPRi.

Response

We thank the reviewer for that helpful suggestion and have added a table that describes the pros and cons of CRISPRc and CRISPRi to the revised manuscript (Table 1).

4. The authors demonstrated that alternative TSSs can impact KRAB-dCas9 gene inhibition. Based on CAGE-seq and H3-K4-me3 Seq data in HT29 cells, it would be nice to demonstrate targeting TSS1 and TSS2 PELO and determine if it leads to effective CRISPRi suppression (Supplementary Figure 3B).

Response

Since the submission of the original manuscript, we analyzed *PELO* expression following CRISPRi mediated suppression of TSS_1 or TSS_2 using quantitative PCR. Unexpectedly, we found targeting of TSS_1 inhibited *PELO* expression in both HT29 and MIAPACA2 cells while *ITGA1* expression was increased in HT29. These observations indicate that predicting the specific effect of CRISPRi on bidirectional promoters likely requires one to test many sgRNA within the region between the two promoters. Since it is clear that further work will be necessary to understand these effects, which may be loci specific, we have removed supplemental figure S3 from the revised manuscript but have added text to describe this complexity (p. 8).

5. In page 7, the authors indicated that they re-evaluated CAGE-seq promoter annotations and found 4.7% of protein-coding genes share bidirectional promoters (Fig. 3D). However, this figure is missing in the supplementary information.

Response

We thank the reviewer for noticing and apologize for the typo we corrected the figure to Fig. 3C. In addition, we have added a more detailed description of bidirectional promoters to the revised manuscript and have also added these new figures (Fig S2C-G).

6. There are a number of typos in the manuscript and mistakes in the figures, and please carefully revise the manuscript and correct them. For example, Supplementary Figure legends 3A and 3B are reversed; Fig 1B might have wrong labels for "Cas9" and "KRAB-dCas9".

Response

We thank the reviewer for noticing these typos. We have carefully reviewed the manuscript and have corrected these typos.

7. The references: referring to the CRISPRi method should include original publications on the topic as reported in Qi et al. 2013; Mali et al. 2013; etc.

Response

We have added the references as suggested by the reviewer.

Reviewer #2:

1. The authors demonstrated that TSS-targeting guides exhibited very similar phenotypes regardless of whether Cas9 or dCas9-KRAB were used for many genes, even for untranslated regions (Figs 1B, 1D, S1B). This is quite a surprising finding, given that the principal mechanism of Cas9 gene inactivation is thought to be by inducing frameshift mutations in protein coding regions. Upon further inspection, I noticed that many of the TSS-targeting guides exhibiting the strongest lethal phenotypes with Cas9 contain low complexity repeats (eg. GCCGCCGCCGCCGCC). This is likely due to how the tiling library guides were chosen with no filtering for multi-cutters or low complexity sequences. The authors should address whether these TSS regions are inherently less complex and therefore more likely to cause non-specific lethality due to multi-site cutting (as described in the prior work cited by the authors). I would strongly recommend to either remove (or at least clearly label in the graphs) all potential multi-cutters and low complexity sequences before performing downstream analysis, as it may be biasing results.

Response

We too were surprised to find that CRISPRc had a very similar effect to CRISPRi in promoter regions. The reviewer suggested that low sgRNA complexity resulting a non-specific cutting and killing effect are the basis of this observation. Although the reviewer raises a potential explanation, we performed several different experiments that provide strong evidence that this is not likely to be the case. Specifically:

- a) We found a similar correlation between CRISPRi and CRISPRc in a positive selection screen that cannot be explained by non-specific killing (6TG resistance Fig. 1B and Table S1).
 - b) When we examined the results from using CRISPRc or CRISPRi to target promoter regions of genes that are not essential for proliferation such as *HPRT1* or *APC*, we failed to find an inhibitory effect on cell proliferation, demonstrating that off target effects induced by low complexity sgRNAs are not found for many genes analyzed.
 - c) Since the submission of our manuscript, a similar observation has been made using enhancer targeting sgRNAs using CRISPRi and CRISPRc (Zhang et al. *Nature Genetics*, 2016, PMID: 26656844).
2. Is there any evidence of these same low-complexity sequences showing increased lethality with dCas9-KRAB by promiscuous multi-targeting of the TSS (or other genomic regions) for many transcripts? Fig 2E addresses perfect matches, but not potential repeats that don't match completely to other sites in the genome (a very common 17-mer or similar).

Response

To address the reviewer's comment, we have re-analyzed our multi alignment targeting analysis using non-perfect alignments (1 mismatch) and did not find an increased anti proliferation effect for sgRNAs that align to multiple genomic loci in KRAB-dCas9. We have added this new analysis to the revised manuscript (Fig. 2SC). In addition, when we used for alignment only partial sgRNA sequences (17 or 18 bp), we found a similar trend (Fig. S2D-E). Together these observations make it highly unlikely that promiscuous multi-targeting of the TSS is responsible for the observed effects and thank the reviewer for their suggestion.

3. Raw count data should be provided in the supplementary data files so that re-normalization can be performed. Additionally, the annotations in Fig. 1C,D (distance from TSS) are not annotated in the supplementary data, and should be added.

As requested by the reviewer, we have added raw data count as a Supplemental table (Supplemental tables S7-S10).

4. Minor point: The finding that CRISPRi screens targeting bidirectional promoters can lead to false-positives is an important finding. I wonder if it is at all possible to identify sgRNAs around bi-directional promoters that preferentially silence one transcript over the other, or if always both genes are strongly impacted (which may be the case due to epigenetic silencing spread by KRAB domain). One way to address this question is to run a RT-PCR for both genes for a number of the sgRNAs in the tiling array? Or can the authors think of a way to addressing this question with their current dataset?

Response

We thank the reviewer for these suggestions. Our current manuscript addresses these concerns by using a combined CRISPRi and CRISPRc approach which is used to identify and flag these genes. The approach suggested by the reviewer may be useful in identifying roles for CRISPRi sgRNA design and will require production of large number of individual sgRNAs and many new experiments that are beyond the scope of this manuscript. We agree however that this point merits further discussion and have added text to the revised manuscript to address these points (p. 8 of the revised manuscript).

Please also see the response to Reviewer #1, points 2 and 4.

Reviewers' Comments:

Reviewer #1:

Remarks to the Author:

The authors have addressed all our concerns, and therefore acceptable for publication.

I would like to point out a few typos which need to be corrected:

Manuscript page 6 line 19: please omit "at"

Supplementary figure legends line 6: "TSS" targeting sgRNAs are colored in blue, not exon (repeated).

Reviewer #2:

Remarks to the Author:

The authors have satisfactorily address the questions and comments raised during review. I feel that the manuscript is suitable for publication with no further modifications.

Detailed responses to reviewers

Reviewer #1:

The authors have addressed all our concerns, and therefore acceptable for publication.

We thank the reviewer for reviewing our manuscript.

I would like to point out a few typos which need to be corrected:

Manuscript page 6 line 19: please omit "at"

We have edited this sentence.

Supplementary figure legends line 6: "TSS" targeting sgRNAs are colored in blue, not exon (repeated).

We have corrected the figure legends.

Reviewer #2:

The authors have satisfactorily address the questions and comments raised during review. I feel that the manuscript is suitable for publication with no further modifications.

We thank the reviewer for reviewing our manuscript.